# Interplay Between 3D Chromatin Architecture and Gene Regulation at the *APOE* Locus Contributes to Alzheimer’s Disease Risk

**DOI:** 10.3390/ijms27010302

**Published:** 2025-12-27

**Authors:** Eun-Gyung Lee, Lesley Leong, Sunny Chen, Jessica Tulloch, Chang-En Yu

**Affiliations:** 1Geriatric Research, Education, and Clinical Center, VA Puget Sound Health Care System, Seattle, WA 98108, USA; lesley.leong@va.gov (L.L.); sunny.chen@va.gov (S.C.); jessica.tulloch@va.gov (J.T.); 2Department of Medicine, University of Washington, Seattle, WA 98195, USA

**Keywords:** Alzheimer’s disease, *APOE* locus-associated genetic variants, gene regulation, 3D genome organization, chromatin interaction, *APOE*, *TOMM40*, *APOC1*, chromosome conformation capture, digital PCR

## Abstract

The ε4 allele of the apolipoprotein E (*APOE*) gene strongly increases Alzheimer’s disease (AD) risk, though its molecular mechanisms remain unclear. AD-associated genetic signals also extend to neighboring genes *TOMM40* and *APOC1*, suggesting a complex cis-regulatory landscape. To investigate chromatin architecture and its impact on gene regulation across this region, we performed chromosome conformation capture in human cell lines and postmortem brain tissues, consistently identifying *TOMM40*–*APOE* and *APOE*–*APOC1* interactions. We further developed a digital PCR assay to quantify *APOE*–*APOC1* interaction strength and measured *APOC1* mRNA via RT-qPCR. Enhanced chromatin interaction correlated with elevated *APOC1* transcription in AD specimens. Genotypic analysis showed that ε3/ε4 carriers had strong chromatin interaction and transcriptional activation, whereas ε4/ε4 homozygotes exhibited minimal chromatin remodeling despite similar *APOC1* expression, suggesting a decoupling of chromatin architecture and transcriptional output. These findings underscore the interplay of AD status, *APOE* genotype, and locus-specific chromatin dynamics in disease susceptibility. Integration of 3D genome topology with transcriptomic profiling offers a framework to study *APOE*-related disorders and supports broader application across neurodegenerative loci for genotype-guided therapy development.

## 1. Introduction

Over the past three decades, the apolipoprotein E gene (*APOE*) and its major variants (ε2, ε3, ε4), defined by two common SNPs (rs429358 and rs7412), have been central to genetic research. These alleles are associated with diverse conditions, including aging/longevity [1,2], Alzheimer’s disease (AD) [3], coronary artery disease [4], diabetes [5], fragile X-associated ataxia [6], glomerulopathy [7], Lewy body dementia [8], metabolic syndrome [9], retinal-related disorders [10], Parkinson’s disease [11], stroke [12], posttraumatic stress disorder [13], and vascular dementia [14]. Although the molecular mechanisms by which *APOE* drives disease pathogenesis remain incompletely defined, its pleiotropic effects underscore a pivotal role in biological pathways that influence disease susceptibility and human health.

Among *APOE*-associated disorders, AD is the most impactful, with the ε4 variant consistently validated as a major risk factor in large, multi-ethnic genome-wide association studies (GWAS) [3,15,16]. In Caucasian populations, the ε4 allele frequency is higher among AD patients (0.36–0.42) than in healthy controls (0.14–0.16) [17,18]. The ε4 allele increases AD risk in a dose-dependent manner and is associated with earlier disease onset [19]. In contrast, the ε2 allele may provide modest protection against AD [20]. Recent genome-wide association studies (GWAS) and next-generation sequencing analyses have identified more than 100 additional genetic loci linked to AD, each conferring modest effect sizes (odds ratios ranging from 1.1 to 1.5) [21,22,23]. Despite these advances, the *APOE* ε4 allele remains the most potent genetic risk factor for AD, with an odds ratio of 3.7 [24], underscoring its central role in disease susceptibility.

Research on *APOE* in AD has traditionally focused on its protein product, apoE. The ε2, ε3, and ε4 variants encode three isoforms (apoE2, apoE3, and apoE4) with distinct affinities for lipoprotein particles and low-density lipoprotein receptors [25], leading to isoform-specific differences in cholesterol levels [26]. These differences have generated multiple hypotheses on how apoE4 protein may increase AD risk, including Aβ aggregation and clearance, tau hyper-phosphorylation, apoE domain interaction, neuroinflammation, neuroprotection, and differential lipidation states [27,28,29,30,31,32]. In addition, emerging protective gene variants such as the *APOE* Christchurch variant have been shown to modulate receptor engagement, enhance microglial responses, and strengthen innate immunity, thereby conferring resilience against AD. Comprehensive reviews of these mechanisms are available [33,34].

However, the presence of apoE4 alone is insufficient to cause AD. Some ε4 carriers who produce apoE4 protein throughout their lives remain cognitively intact into their 90s [35,36], and even homozygous carriers may avoid AD [37]. The impact of ε4 on dementia and mortality diminishes in very old age (90+ years) [38,39]. Interestingly, studies in younger individuals (<30 years) show ε4 carriers performing better on tests of executive function, attention, and memory [40,41,42,43]. These findings suggest that *APOE* ε4 and its E4 protein isoform exert complex, pleiotropic effects that do not fully explain AD etiology, and that additional biologically relevant factors beyond *APOE* itself likely contribute to its strong genetic association.

The GWAS have consistently identified strong genetic associations between SNPs in three adjacent genes within the *APOE* locus—*TOMM40*, *APOE*, and *APOC1*—and AD [44,45]. This shared genetic variance also correlates with coronary artery disease [46] and longevity [2,47]. Such a profile could arise from one of two possibilities: either *APOE* alone drives the association, with nearby signals acting as proxies due to strong linkage disequilibrium [48], or the combined impact of all three genes contributes collectively to AD and related traits. Although distinguishing between these scenarios remains challenging, the latter offers a novel perspective, suggesting that multiple genes within the *APOE* locus may act together to shape the biological basis of AD. This concept aligns with the view that *APOE* functions as a risk factor rather than a causative gene. The development of AD is likely contingent not only on the ε4 allele but also on other genetic variants in neighboring genes. Different allele combinations create diverse ε4-linked haplotypes with varying effects on the coregulation of these genes, consequently leading to distinct AD risk outcomes. Thus, this second scenario has strong scientific merit and highlights the need to explore additional molecular mechanisms, with epigenetics emerging as a notable contender.

Epigenetic mechanisms add a dynamic layer of regulatory complexity, mediating the interface between genotype and phenotype beyond the static DNA sequence. Covalent modifications, such as DNA methylation and post-translational histone modifications, drive chromatin remodeling, influencing nucleosome positioning as well as higher-order chromatin organization. These processes shape genome architecture, including nucleosome clusters, chromatin interaction (CI) loops, topologically associating domains, and A/B compartments [49,50,51]. Within the three-dimensional (3D) nuclear landscape, CI loops connect regulatory elements such as enhancers and silencers to their target genes, impacting transcription and phenotypic variation in mammals [52,53]. CIs are essential for initiating transcription and fine-tuning gene expression, providing a foundation for precise genomic regulation [54].

In this context, CI loops provide a framework for understanding distal regulatory architecture at the *APOE* locus in AD. *TOMM40*, *APOE*, and *APOC1* may be coregulated through intra-chromosomal CIs, collectively shaping susceptibility. Variants can disrupt looping, alter chromatin structure, and impair transcription, driving heterogeneity in AD risk. Consistent with current models, *APOE* allelic variants (e.g., ε2/ε3/ε4) modulate AD risk, while polymorphisms in adjacent genes such as *TOMM40* and *APOC1* also exhibit strong disease associations. These findings support a model where CIs integrate regulatory effects with significant implications for AD pathogenesis.

Advances in 3D genomics, particularly chromatin conformation capture (3C) and related methodologies, have revealed long-range interactions between genes and distal regulatory elements, offering key insights into spatial gene regulation. Using 3C-based analyses, this study examines chromatin interactions at the *APOE* locus to clarify their role in transcriptional regulation and assess their contribution to AD susceptibility.

## 2. Results

### 2.1. Design a Chromosome Conformation Capture Assay Targeting the APOE Locus

To investigate how chromatin architecture at the *APOE* locus may influence gene regulation and contribute to AD susceptibility, we employed the chromosome conformation capture (3C) assay to identify locus-specific CIs and evaluate their regulatory impact in the context of AD risk. This 3C assay allows identification of genuine physical connections between two separated chromatin regions, designated as the anchor-target pair. Prior to conducting the 3C assay, we carried out a bioinformatic analysis to identify candidate anchor regions within the *APOE* locus. Utilizing publicly available resources like FORGEdb [55] and TADKB [56], which provide low-resolution CI maps of the *APOE* locus, we selected two regions as anchor sites for our study: the *APOE* promoter and the *APOE* exon 4 CpG island (CGI) [57]. Figure 1 illustrates the publicly available datasets characterizing the chromatin architecture and regulatory landscape of the *APOE* locus, integrating genomic interaction profiles, gene annotations, and histone modification patterns.

### 2.2. Identification and Quantification of Cell Type-Specific Chromatin Interactions at the APOE Locus in Human Cell Lines

As an initial proof-of-concept for chromatin interactions, we performed the 3C assay on several human cell lines, including hepatocyte-like (HepG2), astrocyte-like (U87), microglia-like (HMC3), and neuron-like (SH-SY5Y) cells. Using the *APOE* promoter as an anchor, we identified CI with a region spanning introns 2 to 3 (IVS2-3) of *TOMM40* (see Figure 1C, red lines). This interaction was observed in both U87 and HMC3 cells, but not in SH-SY5Y cells (Appendix A). This finding aligns with Nuytemans et al. [58], who reported a similar interaction between the *APOE* promoter and a similar region (IVS2-5) of *TOMM40* in primary human astrocytes and microglia (see Figure 1C, purple lines). Additionally, this target site on *TOMM40* overlaps with H3K4Me1 (see Figure 1D), a well-defined histone mark for regulatory elements, and contains SNPs (e.g., rs2075650) associated with AD risk [59]. These findings suggest that this *TOMM40* region may function as a regulatory element—either an enhancer or silencer—whose activity is modified by genetic variants to influence *APOE* expression in a cell type-specific manner. When using the *APOE* exon 4 CGI as an anchor, we detected a CI with the *APOC1* promoter region (Figure 1C, green lines) across all tested cell lines (Appendix A). The consistent presence of this target-anchor pair across various cell types suggests a fundamental CI between *APOE* and *APOC1*.

To precisely quantify CI strength, we developed a digital PCR–based assay termed CIS-dPCR (CI Strength digital PCR; see Appendix A). This assay enables absolute measurement of chimeric anchor–target DNA fragments within complex genomic pools through sample partitioning and single-molecule detection on a digital PCR platform. CI strength values are normalized to defined cell counts or genomic DNA input, permitting direct comparison across biological variables such as genotype, cell type, and disease stage. Details of the normalization procedure are provided in the Section 4.

To validate this CIS-dPCR assay, we directed our investigation toward the CI between *APOE* and *APOC1*, given that this anchor-target pair encompasses the ε2/ε3/ε4 allelic variants of *APOE*. We hypothesized that these variants may alter the frequency or strength of this interaction. We quantified the CIs strength between the CGI of *APOE* exon 4 and the *APOC1* promoter across four human cell lines: HepG2, U87, HMC3, and SH-SY5Y. Their strengths varied substantially across cell types (Figure 2A), indicating cell type-dependent chromatin architecture. Given the involvement of the *APOC1* promoter in this interaction, we concurrently assessed *APOC1* expression levels to evaluate potential transcriptional consequences. *APOC1* mRNA levels were quantified by RT-qPCR using a TaqMan probe set, normalized to *ACTB* as an endogenous control. ∆∆Ct values were calculated relative to SH-SY5Y cells, which served as the reference baseline (expression level set to 1). *APOC1* transcriptional levels also varied in a cell type-specific manner. Notably, cell lines exhibiting stronger *APOE–APOC1* CIs also demonstrated elevated *APOC1* mRNA levels (Figure 2B), supporting a mechanistic link between CI and gene expression. These findings further suggest that locus-specific chromatin architecture influences the cell type-specific transcriptional regulation of *APOC1* within the *APOE* genomic region.

### 2.3. Chromatin Interaction Strength and Gene Expression Profiles in Human Postmortem Brain Tissues

Building on proof-of-concept findings in cell line models, we expanded our analysis to human frontal lobe tissue obtained from postmortem brain (PMB) specimens to evaluate the presence of the *APOE*–*APOC1* CI within the central nervous system. Demographics of the PMB samples are listed in Table 1.

Using the same 3C protocol, we consistently detected this CI across all PMB samples. To assess whether this CI is altered in AD or modulated by the *APOE* ε4 allele, we quantified CI strength and *APOC1* mRNA expression in PMB samples via CIS-dPCR (Appendix A) and RT-qPCR. Stratification by disease status revealed a significant increase in CI strength in AD samples relative to controls (Ctrl) (*p* < 0.05), indicating enhanced physical proximity between the *APOE* exon 4 CGI and the *APOC1* promoter in AD brains (Figure 3A).

Concurrently, *APOC1* mRNA levels were significantly elevated in AD samples (*p* < 0.005), suggesting that increased locus-specific CI may contribute to transcriptional upregulation in the disease context. Correlation analysis revealed distinct regulatory patterns: in Ctrl samples, CI strength and *APOC1* expression were inversely correlated (r = −0.56, *p* = 0.09), whereas a positive, though nonsignificant, correlation was observed in AD samples (r = 0.39, *p* = 0.15), implying disease-associated alterations in the relationship between chromatin architecture and gene expression. Further stratification by *APOE* genotype demonstrated significant variation in CI strength, with ε3/ε4 carriers exhibiting higher interaction levels than ε3/ε3 individuals (*p* < 0.05; Figure 3B), indicating genotype-dependent modulation of chromatin structure. Although *APOC1* mRNA levels showed a genotype-associated trend, differences did not reach statistical significance (*p* > 0.05). Notably, genotype-stratified correlation analysis revealed a moderate positive association between CI strength and *APOC1* expression in ε3/ε4 carriers (r = 0.61, *p* = 0.06), while no significant correlations were detected in ε3/ε3 or ε4/ε4 individuals. These findings suggest that CI dynamics at the *APOE*–*APOC1* locus are influenced by both disease status and *APOE* genotype, potentially contributing to dysregulated *APOC1* transcription in AD.

To evaluate the individual and interactive effects of *APOE* genotype and AD status on CI strength and *APOC1* transcript levels, we conducted regression analyses across stratified sample groups (Table 2). AD status alone was moderately associated with increased CI strength (β = 0.247), with a more pronounced elevation observed in individuals harboring the *APOE* ε3/ε4 genotype (β = 0.408). In contrast, the ε4/ε4 genotype exerted minimal influence (β = 0.024). The combined presence of AD and the ε3/ε4 genotype yielded the strongest association with CI strength (β = 0.427). Regarding *APOC1* mRNA expression, AD status alone was weakly associated with increased transcript levels (β = 0.055). Carriers of the ε3/ε4 genotype exhibited elevated expression (β = 0.285), with the highest levels detected in ε4/ε4 individuals (β = 0.570). The co-occurrence of AD and the ε3/ε4 genotype produced the greatest increase in *APOC1* expression (β = 0.651), suggesting a genotype-dependent modulation that may be further amplified in the context of AD. Regression estimates for the AD + ε4/ε4 subgroup were unavailable, likely due to limited sample size or reduced model stability. Collectively, these findings delineate a complex regulatory landscape in which *APOE* genotype and AD status interact to influence chromatin architecture and transcriptional activity at the *APOE* locus.

To assess whether additional genetic variants within the *APOE*–*APOC1* CI domain modulate CI dynamics, we examined the regulatory influence of a common insertion (in)/deletion (del) polymorphism (rs11568822) located in the *APOC1* promoter region. CI strength and *APOC1* mRNA expression were quantified in PMB samples stratified by rs11568822 genotype (del/del, in/del, in/in). CI strength differed significantly among genotypic groups (*p* < 0.05), with heterozygous in/del individuals exhibiting the highest interaction levels (Appendix A). These findings suggest that the insertion allele may enhance locus-specific chromatin proximity, potentially facilitating regulatory contacts within the *APOE*–*APOC1* interaction. *APOC1* transcript levels demonstrated genotype-dependent variation, although differences did not reach statistical significance (*p* > 0.09; Appendix A). Nonetheless, the observed expression trend supports a putative modulatory role for rs11568822 in regulating *APOC1* transcription. These findings implicate rs11568822 as a candidate effector of CI looping and transcriptional activity, reinforcing the hypothesis that genetic variation within the *APOE* locus contributes to 3D genome organization and downstream regulatory outcomes relevant to AD pathogenesis. Furthermore, the concordant patterns of CI strength and *APOC1* expression across stratifications by both *APOE* genotype and *APOC1* rs11568822 allelic status suggest strong linkage disequilibrium between these loci, with frequent co-occurrence of the ε4 and insertion alleles on the same haplotype.

### 2.4. Impact of Regional DNA Methylation on Chromatin Interactions and Gene Expression

Prompted by evidence of elevated DNA methylation (DNAm) at the CGI within *APOE* exon 4, along with reported differential methylation between AD and control samples in human frontal lobe tissue [60], we investigated whether DNAm modulates the locus-specific chromatin architecture. Using PMB tissue, we assessed the relationship between *APOE*–*APOC1* CI strength and DNAm levels at differentially methylated region I (DMR I) [60]. Stratification by disease status revealed no significant difference in DNAm levels between AD and control samples (*p* = 0.383), indicating that global methylation at this locus is not altered in the disease state (Figure 4A). However, correlation analyses revealed condition-specific associations: in control samples, DNAm levels were significantly positively correlated with CI strength (r = 0.65, *p* = 0.04), suggesting a stabilizing role for DNAm in maintaining chromatin architecture under non-pathological conditions. In contrast, AD samples exhibited a non-significant inverse trend (r = −0.35, *p* = 0.20), raising the possibility of epigenetic–structural decoupling in the context of disease. Further stratification by *APOE* genotype revealed no significant differences in DNAm levels across genotype groups (*p* > 0.49; Figure 4B). Nonetheless, genotype-specific correlations with CI strength were observed. Notably, ε3/ε3 carriers exhibited a significant negative correlation (r = −0.69, *p* = 0.03), indicating that reduced methylation at DMR I may be associated with increased CI in this subgroup. While no significant correlations were detected in ε3/ε4 (r = −0.13, *p* = 0.73) or ε4/ε4 (r = 0.27, *p* = 0.66) individuals.

To further interrogate whether DNAm mediates the interactive effects of disease status and genetic variation at the *APOE* locus on *APOE*–*APOC1* CI strength or *APOC1* gene expression, we performed causal mediation analyses (Table 3). Following adjustment for relevant covariates, the average causal mediation effect (ACME) attributable to DNAm was minimal for both molecular outcomes, with wide confidence intervals and non-significant *p*-values, indicating limited evidence for DNAm-mediated pathways. In contrast, the average direct effect (ADE) on CI strength approached statistical significance (*p* = 0.066), and the total effect was significant (*p* = 0.042), suggesting a robust overall association that is largely independent of DNAm. For *APOC1* transcript levels, neither the direct nor total effects reached statistical significance, further supporting the absence of a methylation-mediated mechanism. These findings collectively indicate that DNAm does not substantially mediate the impact of disease status and genetic variation on CI strength or *APOC1* mRNA expression. Instead, they highlight the complex, multifactorial nature of epigenetic regulation and 3D chromatin architecture, which may be shaped by neurodegenerative pathology and *APOE* locus-associated genetic context through mechanisms beyond methylation.

## 3. Discussion

This study demonstrates that locus-specific chromatin interactions within the *APOE* locus define regulatory architecture and establish CIS-dPCR as a sensitive tool for mapping functional 3D genome organization. We show that *APOE*–*APOC1* chromatin interactions are dynamically shaped by AD status and *APOE* genotype, while DNA methylation exerts only limited influence. These interactions are directly implicated in transcriptional regulation, with the *APOE* CGI functioning as a noncanonical enhancer of *APOC1* expression. Together, our findings highlight chromatin interactions as key regulators of transcriptional output and as promising therapeutic targets.

The *APOE* gene, particularly the ε4 allele, plays a major role in the risk and progression of AD. Although the molecular mechanisms underlying this association remain incompletely understood, *APOE* genotyping is gaining clinical significance for risk assessment, early detection, and the development of potential therapeutic strategies. As research continues to advance, a better understanding of the role of the *APOE* locus in AD may pave the way for more effective prevention and treatment strategies for this devastating neurodegenerative disorder.

Convergent lines of evidence indicate that local ancestry at the *APOE* locus shapes distinct cis-regulatory landscapes across populations, modulating AD risk via mechanisms beyond the influence of canonical *APOE* genotypes. Studies have demonstrated that haplotypes of African versus European origin within the *APOE* region exhibit distinct chromatin accessibility profiles and differential expression of the *APOE* ε4 allele, with African-derived haplotypes associated with attenuated *APOE* ε4 expression and reduced AD [61,62]. These ancestry-specific effects are likely mediated by regulatory variants in linkage disequilibrium with *APOE*, including elements that modulate transcription factor binding and epigenetic marks across the *TOMM40*–*APOE*–*APOC1* gene cluster [63]. Such findings underscore the importance of incorporating local ancestry and cis-regulatory context into models of *APOE*-mediated disease susceptibility, particularly in defined populations.

The interplay between chromatin remodeling and 3D genome organization underlies essential epigenetic mechanisms for precise gene regulation. Remodeling mediated by DNA methylation and histone modifications modulates chromatin accessibility, acting as transcriptional switches responsive to cellular signals [64]. Spatial chromatin architecture further enables long-range enhancer–promoter interactions, clustering co-regulated genes and concentrating transcriptional machinery [65]. GWAS have consistently identified AD-associated variants within the *TOMM40–APOE–APOC1* cluster that may alter spatial architecture and influence locus-specific transcriptional dynamics [44,45]. Mapping its 3D configuration provides insight into regulatory mechanisms underlying AD pathogenesis [63].

3C-based analysis revealed both cell type-specific and conserved regulatory architectures at the *APOE* locus, underscoring potential implications for AD risk. A glia-specific interaction between the *APOE* promoter and the *TOMM40* IVS2-3 region was observed in astrocyte-like (U87) and microglia-like (HMC3) cells, but not in neuron-like (SH-SY5Y) cells, consistent with prior reports in primary human glial cells [58] and supporting restricted *APOE* expression under physiological conditions and is induced in neurons only under stress [66]. The *TOMM40* region harbors enhancer-associated marks (H3K4me1) and AD-linked variants such as rs2075650, suggesting possible genetic modulation of regulatory activity. In contrast, the consistent interaction between the *APOE* exon 4 CGI and the *APOC1* promoter across all cell types points to a conserved cis-regulatory mechanism. Together, these findings highlight the complexity of chromatin topology at the *APOE* locus and support a model in which 3D genome organization contributes to both cell type-specific regulation and AD susceptibility.

The CIS-dPCR and RT-qPCR assays across multiple human cell lines revealed a strong correspondence between CI strength and gene expression, highlighting regulatory architecture at the *APOE–APOC1* locus. Interaction strength varied across HepG2, U87, HMC3, and SH-SY5Y cells, indicating cell type-dependent chromatin looping between the *APOE* exon 4 CGI and the *APOC1* promoter. Cell lines with stronger CIs also exhibited elevated *APOC1* transcript levels, suggesting that 3D genome configuration influences transcriptional output. These findings underscore the biological significance of *APOE–APOC1* contacts and point to locus-specific looping as an epigenetic axis modulated by cell identity. CIS-dPCR thus emerges as a sensitive, scalable tool for quantifying CIs and mapping functional regulatory landscapes, while providing a framework for future studies on how genetic variation at the *APOE* locus shapes chromatin organization and AD-linked gene networks.

Detection of *APOE–APOC1* CIs in human frontal lobe tissues enabled assessment of their physiological relevance in the brain. In all examined PMB samples, we consistently observed interactions between the *APOE* exon 4 CGI and the *APOC1* promoter, mirroring those in diverse cell lines. These findings confirm the reproducibility of this interaction in native brain tissue, suggesting that locus-specific chromatin looping is preserved in vivo and contributes to regulatory architecture in the central nervous system.

Our findings show that chromatin architecture at the *APOE–APOC1* locus is dynamically modulated by both AD status and *APOE* genotype, with implications for transcriptional regulation in disease. Enhanced CI and elevated *APOC1* expression in AD brains suggest locus-specific reconfiguration of 3D genome organization that facilitates transcriptional upregulation. Notably, the correlation between CI strength and gene expression shifts from inverse in controls to positive in AD, indicating disease-associated rewiring of regulatory logic. Genotype-stratified analyses highlight the distinct profile of ε3/ε4 carriers, who exhibited stronger looping and moderate correlation with *APOC1* expression, pointing to genotype-specific effects. The combined presence of AD and ε3/ε4 genotype produced the most pronounced architectural and transcriptional changes, suggesting a synergistic effect. Our findings support a model in which genes within the *APOE* locus jointly contribute to AD susceptibility through 3D chromatin remodeling (Appendix A). Genetic variants and AD status reshape local chromatin architecture, modulating promoter–enhancer interactions and transcriptional output from regulatory elements such as *APOC1* and *APOE*. These results emphasize that locus-wide regulatory dynamics, rather than single-gene effects, underlie *APOE*-associated risk and highlight allele-specific chromatin organization as a key factor in AD susceptibility. Multiple studies demonstrate that 3D chromatin remodeling is causally linked to transcriptional regulation across development, aging, and disease. Structural proteins such as CTCF, cohesin, and Polycomb complexes establish loops and domains, and perturbation studies show that their disruption directly alters gene expression [67,68,69].

We identified the *APOC1* promoter variant rs11568822 as a potential modulator of chromatin architecture. This in/del polymorphism may influence *APOE–APOC1* interaction in a genotype-specific manner, as individuals with the heterozygous in/del genotype showed stronger CI and elevated *APOC1* expression, suggesting that the insertion allele stabilizes enhancer–promoter contacts with regulatory consequences for AD risk. Its frequent co-occurrence with the ε4 allele indicates a shared haplotypic framework that may jointly shape chromatin topology and transcriptional output at this locus.

Our findings indicate that DNA methylation at the *APOE* locus exerts only a limited influence on chromatin interaction and transcriptional regulation. Overall DNAm levels did not differ significantly between AD and control brains, but condition-specific correlations suggested divergent dynamics. In controls, DNAm positively correlated with CI strength, implying stabilization of looping under non-pathological conditions. In contrast, AD brains showed a non-significant inverse trend, raising the possibility of epigenetic–structural uncoupling. Genotype-stratified analyses added further nuance. In ε3/ε3 carriers, lower methylation was significantly associated with stronger CI, suggesting a fine-tuning role. No significant correlations were observed in ε3/ε4 or ε4/ε4 individuals, which may reflect genotype-specific susceptibility or limited statistical power. Mediation analyses further showed minimal causal effects of DNAm on CI strength or *APOC1* expression, whereas direct effects approached significance, indicating that disease status and genotype impact chromatin architecture largely independent of methylation. Together, these findings highlight the complexity of epigenetic regulation and 3D chromatin organization, shaped by genetic, epigenetic, and pathological factors. While DNAm at the *APOE* locus may contribute to local chromatin states, it is not a primary mediator of genotype- or disease-associated changes in CI and gene expression.

The *APOE* CGI functions as a noncanonical enhancer that regulates *APOC1* expression through chromatin interaction. We observed a specific loop between the *APOE* CGI and the *APOC1* promoter, suggesting enhancer-like activity despite the absence of canonical histone marks such as H3K4me1, H3K27ac, and p300. Unlike typical enhancers, this region is hypermethylated [60], raising the possibility that hypermethylated CGIs can act as regulatory elements under certain contexts [70,71]. Our previous work confirmed its enhancer activity at the *APOE* locus via reporter assays [57], consistent with Bell et al. [72], who described orphan CGIs as enhancer CGIs (ECGIs) defined by broad transcriptional activity, evolutionary conservation, dynamic methylation, and extensive chromatin contacts. In line with this framework, the *APOE* CGI–*APOC1* promoter interaction supports its role as a robust ECGI, contributing to *APOC1* regulation through a CpG-rich, transcriptionally active, and dynamically methylated chromatin landscape.

The *TOMM40*–*APOE* chromatin interaction represents a potential regulatory mechanism influencing AD risk and longevity. Although incompletely characterized, this interaction has been prioritized for future investigation given its potential role in modulating AD risk at the *APOE* locus. This CI encompasses the *APOE* promoter, suggesting a regulatory function in *APOE* transcription. Supporting evidence includes a candidate enhancer within *TOMM40* intron 2–3 that harbors SNP rs2075650, a variant strongly associated with AD susceptibility and longevity. Allelic variation at rs2075650 may influence CI strength and *APOE* expression. In combination with *APOE* ε2/ε3/ε4 alleles, rs2075650 contributes to distinct haplotypes across the locus, which may differentially shape cis-regulatory architecture and transcriptional output, thereby affecting AD risk and longevity outcomes.

Collectively, this study provides compelling evidence that the three genes within the *APOE* locus—*TOMM40*, *APOE*, and *APOC1*—are coregulated within a 3D genomic architecture. From a biological perspective, both *TOMM40* and *APOC1*, in addition to *APOE*, have been implicated in AD. *TOMM40* encodes Tom40, a critical component of the mitochondrial translocase complex responsible for importing nuclear-encoded proteins into mitochondria. Given that mitochondrial dysfunction is a hallmark of AD pathology [73], reduced *TOMM40* expression impairs mitochondrial integrity [74], leading to metabolic dysregulation, oxidative stress, and mitochondrial damage—processes closely linked to AD progression [75]. *APOC1* encodes apolipoprotein C1, which is involved in lipid metabolism [76]. and innate immune responses [77,78]. Elevated apoC1 levels in the brain, along with *APOC1* genetic variants associated with increased AD risk [79], underscore its relevance to neurobiology and AD pathogenesis. Notably, an upregulation of *TOMM40*, *APOE*, and *APOC1* has been observed in human cellular models subjected to oxidative stress and in PMB tissues from AD patients [80]. Together, these findings suggest that multiple genes within the *APOE* locus may act in concert to modulate AD susceptibility. The demonstration of physical CI and coordinated expression among *TOMM40*, *APOE*, and *APOC1* reinforces the concept of locus-wide coregulation, with potential synergistic effects on AD-related molecular pathways.

This study delineates a novel therapeutic avenue for AD by targeting CI-mediated enhancer and silencer elements at the *APOE* locus. These cis-regulatory motifs represent viable, druggable targets for precise, locus-specific modulation of gene expression. Fine-tuning their activity may help restore transcriptional homeostasis and correct gene expression abnormalities underlying AD pathogenesis. Unlike promoter regions, which are structurally complex and central to transcriptional initiation, enhancer and silencer motifs are less intricate, more modular, and thus more amenable to intervention. This tractability supports the design of DNA mimetics with allele-specific sequences that can modulate transcriptional output by recruiting or sequestering CI-associated proteins to promote loop formation or sequestering them to inhibit looping. Their compatibility with non-invasive delivery methods circumvents the need for complex genetic manipulation [81]. Overall, this strategy enables dynamic, locus-specific regulation of disease-relevant targets across quantitative, qualitative, and temporal dimensions.

Enhancer and silencer elements within the *APOE* locus provide a basis for designing DNA mimetics to modulate transcription. These constructs, used individually or in multiplexed formats, can selectively activate or repress gene expression to mitigate AD-associated dysfunctions. Because *TOMM40*, *APOE*, and *APOC1* are coordinately regulated, multiplexed mimetics are well suited to fine-tune transcriptional output across the locus. Leveraging chromatin-based mechanisms, this strategy offers locus-wide regulatory control to address complex transcriptional dysregulation in AD pathogenesis.

## 4. Materials and Methods

### 4.1. Human Postmortem Brain (PMB) and Cell Lines

Our study uses de-identified human PMB tissues obtained from the University of Washington (UW) Alzheimer’s Disease Research Center, following approval by the institutional review board of the VA Puget Sound Health Care System (MIRB#00331). Because the study involves de-identified human PMB tissue, it is not considered human subjects research under U.S. federal regulations and is therefore exempt from IRB oversight at the VA Puget Sound Health Care System.

AD patient diagnosis was confirmed postmortem by neuropathological evaluation. Clinically normal subjects were cognitively intact volunteers over the age of 65, with no clinical diagnosis of AD and no evidence of AD neuropathology at autopsy. Postmortem frontal lobe samples were obtained from the tissues that were rapidly frozen at autopsy (<10 h after death) and stored −80 °C until use. Hepatocytoma HepG2 and glioblastoma U87 MG cells (ATCC, Manassas, VA, USA) were grown in 89% Dulbecco’s modified Eagle’s medium (DMEM) (Gibco, Grand Island, NY, USA); neuroblastoma SH-SY5Y cells (ATCC) were grown in 89% DMEM with F12 (Gibco); microglia HMC3 cells (ATCC) were grown in 89% Eagle’s Minimum Essential Medium (EMEM) (ATCC). These media were supplemented with 10% fetal bovine serum (FBS) (Gibco). All cell cultures were supplemented with 1% penicillin/streptomycin (Invitrogen, Carlsbad, CA, USA) and incubated at 37 °C in a 5% CO_2_ atmosphere.

### 4.2. DNA/RNA Extraction and Genotyping

Cell line genomic DNA was extracted using the QIAmp DNA Blood Mini Kit (Qiagen, Hilden, Germany) and RNA was extracted using the AllPrep DNA/RNA Mini Kit (Qiagen) according to the manufacturer’s protocols. Genomic DNA and RNA were isolated from frozen PMB using the AllPrep DNA/RNA Mini Kit (Qiagen). Nucleic acid concentrations were measured by NanoPhotometer (Implen, Westlake Village, CA, USA), and samples were stored at −20 °C prior to use. SNPs were genotyped using TaqMan allelic discrimination assays (ThermoFisher, Waltham, MA, USA) according to the manufacturers’ protocols: rs429358 (C_3084793_20) and rs7412 (C_904973_10). A small fragment enclosing the *APOC1* SNP (rs11568822) region was amplified by a standard Hot Start PCR with a primer pair: Ch19_50109453F (5′-ATTCCCCGAACGAATAAACC) and Ch19_50109512R (5′-AGCCGCAGACAAAATTCCT) (Integrated DNA Technologies, Coralville, IA, USA). The PCR fragment was digested with the enzyme HpaI (cut site GTTˆAAC) and analyzed for the fragment size difference between insertion (CGTT) and deletion using QIAxcel with a DNA High-Resolution Cartridge (Qiagen).

### 4.3. Chromosome Conformation Capture (3C)

Step 1: Cross-linking and nuclei isolation. Tissue-cultured cells were filtered through a 40 µm cell strainer and counted to 5–10 million for 3C preparation. PMB tissues (50 mg) were homogenized in Hank’s balanced salt solution (ThermoFisher). These cells were fixed with 2% formaldehyde in isolation buffer (10% FBS in PBS) for 10 min at room temperature to cross-link proteins to DNA through formation of covalent bonds. The cross-linking reaction was quenched with 0.13 M glycine solution and centrifuged at 4 °C. Cell pellets were resuspended in lysis buffer (50 mM Tris-HCl, pH 7.5, 0.5% NP-40, 1% Triton X-100, 150 mM NaCl, 5 mM EDTA, 1× protease inhibitor) and incubated on ice for 20 min. Intact nuclei were pelleted at 2500× *g* for 10 min at 4 °C.

Step 2: Restriction enzyme digestion. The nuclei pellet was conditioned with 0.3% SDS in restriction enzyme digest buffer at 37 °C for 1 h, followed by 2.5% Triton X-100 at 37 °C for 1 h with shaking. Chromatin was then digested with 450 U of restriction enzyme. Digestion was performed under optimized buffer conditions with extended incubation (30 h) and shaking. Restriction sites are expected to be efficiently cleaved unless protected by protein binding (e.g., transcription factors recruited by enhancers). The restriction enzyme was then inactivated.

Step 3: Intramolecular ligation. The digested sample was conditioned with 0.8% SDS at 37 °C for 30 min, then incubated with 10% Triton in ligation buffer (40 mM Tris-HCl, pH 7.8, 10 mM MgCl_2_, 10 mM DTT, 0.5 mM ATP) at 37 °C/2 h with shaking. The sample was centrifuged, leaving 500 µL of supernatant. To this, 190 U of T4 DNA ligase and 3 µL of 100 mM ATP were added, followed by overnight incubation at 16 °C with shaking.

Step 4: Reverse cross-link and DNA purification. The ligated DNA was reverse cross-linked with 100 µg proteinase K at 50 °C for 1 h, then incubated overnight at 65 °C. The samples were treated with 300 µg RNase A at 37 °C for 30 min. The 3C-generated DNA was purified using phenol/chloroform extraction and sodium acetate/ethanol precipitation. The DNA pellet was repurified with a QIAquick PCR purification kit (Qiagen).

Step 5: Validation of 3C-generated CIs using PCR and Sanger Sequencing. For each 20 µL PCR reaction, 25 ng of 3C-generated template DNA was used with the HotStart Taq Plus Master mix kit (Qiagen). Primers targeting the 3C product were designed and synthesized. The thermocycling profile included 10 min at 95 °C, followed by 35 cycles of 20 s at 95 °C, 1 min at 55 °C, and 3 min at 68 °C. After ExoSap-IT treatment (Affymetrix, San Diego, CA, USA) to remove residual primers and nucleotides, a nested PCR was performed using a 1/20 dilution of the first PCR reaction in the HotStart Taq Master mix (Qiagen). The thermocycling profile consisted of an initial denaturation 10 min at 95 °C, followed by 30 cycles of 20 s at 95 °C, 1 min at 55 °C, and 1 min at 70 °C. PCR products were then inspected using a capillary gel electrophoresis system (QiAxcel, ThermoFisher). Single-band PCR products were directly sequenced, while multiple-band products were separated on an agarose gel to extract each band prior to sequencing. Sanger sequencing was performed using BigDye v3.1 terminator cycle sequencing kits, and the sequence data was analyzed on a SeqStudio Genetic Analyzer (ThermoFisher). Nucleotide variants were identified by sequence alignment using SEQUENCHER software v5.4.6 (Gene Codes Corp., Ann Arbor, MI, USA), and original electropherograms were manually inspected for accuracy. Information on the primers is listed in Appendix A. Reciprocal interactions within the *APOE* locus were validated using 3C with various restriction enzymes (4-base and 6-base cutters). Multiple regions at *TOMM40* IVS2–6 interacted with the same *APOE* promoter region, generating restriction-site-dependent PCR fragments of varying sizes. These results confirm the reciprocal chromatin interactions the two regions.

### 4.4. CI Strength-Digital PCR (CIS-dPCR)

The dPCR assay was developed to measure the CI strength between the *APOE* CGI and *APOC1* promoter. We used TaqMan-based PCR assays to quantify this interaction pair. Fluorescent-tagged probes and primers were designed using Primer Quest tool (Integrated DNA Technology, Coralville, IA, USA). The dPCR reaction was performed using the Absolute Q Digital PCR System (ThermoFisher) with a 10 µL reaction mixture, including 10 ng of 3C-generated template DNA, 2 µL of 5× Absolute Q Universal DNA Digital PCR Master Mix (ThermoFisher), and 1 µL of 10× digital PCR assay comprising 8 µM of each primer and 4 µM probe. The thermal cycling profile was set to an initial denaturation at 96 °C/10 min and 96 °C/5 s–60 °C/15 s (40 cycles). Fluorescence signals were scanned and converted into partitions on the dPCR plate for each target sample using the QuantStudio Absolute Q Digital PCR Software 6 (ThermoFisher). These partitions were then normalized with internal and external control counts to obtain the absolute quantifications of the interaction anchor–target pair for each sample. To normalize initial DNA input, 10% of the cross-linked nuclei extract was aliquoted prior to 3C processing and independently reverse cross-linked and purified. The Hemoglobin Subunit Beta (*HBB*) gene [82], which shows no copy-number variation in the human genome, served as the internal reference. To account for DNA loss during multi-step processing, a fixed amount of exogenous Green Fluorescent Protein (*GFP*) DNA (188 bp) [83] was spiked into the nuclei solution as an external reference. A multiplex dPCR assay was then developed to simultaneously detect the anchor–target pair and *GFP* spike-in in each reaction, while the internal *HBB* control was measured in a separate reaction. These normalization strategies reduced technical variability and improved measurement accuracy of the chromatin interactions. For assay reproducibility, two independent 3C libraries and three to four dPCR replicates were performed for each sample. CIS-dPCR workflow and associated control normalization strategy are illustrated in Appendix A. Information on the primers and probes is listed in Appendix A.

### 4.5. Reverse Transcriptase Reaction and Quantitative PCR (RT-qPCR)

RT-qPCR assay was used to measure *APOC1* gene expression. Total RNA (100 ng) was used for each 20 µL RT reaction, and cDNA synthesis was performed using the PrimeScript RT Reagent Kit (Takara Bio USA, Mountain View, CA, USA). The resulting cDNA was diluted 20 times for qPCR, and expression levels were measured using TaqMan assays in QuantStudio 5 (ThermoFisher, Waltham, MA, USA). Each 10 µL qPCR reaction contained a fixed RNA input (5 ng), 0.5 µL of the 20× TaqMan assay, and 5 µL of 2× TaqMan Universal PCR Reaction Mix (Applied Biosystems, ThermoFisher). The thermal cycling program consisted of 2 min at 50 °C, 10 min at 95 °C, and then 40 cycles of 15 s at 95 °C and 1 min at 60 °C. Gene expression was assessed by triplicate qPCR assays using two independently prepared reverse transcription (RT) reactions. *APOC1* transcription was quantified using the RT-qPCR ΔΔCt method. To control RNA quantity, *ACTB* mRNA was measured as an internal control for each PMB sample. Normalized ΔCt values were obtained as: mean *ACTB* Ct (triplicate) − Ct (*APOC1*). In this setting, ΔCt values were near or below zero, reflecting higher *ACTB* mRNA levels compared to *APOC1*; larger ΔCt values indicate higher *APOC1* RNA abundance. ΔΔCt values were then derived relative to a reference PMB sample (ΔΔCt = ΔCt [sample] − ΔCt [reference]). Thus, normalization was performed against *ACTB*, with higher ΔΔCt values indicating increased *APOC1* mRNA abundance. Details of the gene-specific TaqMan assays are provided in Appendix A.

### 4.6. Bisulfite Pyrosequencing

DNA methylation levels of the human PMB samples were quantified by pyrosequencing. Detailed procedures have been described in our prior published works [60]. Briefly, genomic DNA (500 ng) was bisulfite converted using the EpiTect Bisulfite Kit (Qiagen). To evaluate the methylation status of the *APOE*’s DMR I, we designed pyrosequencing assays to cover 27 CpG sites (CpG 11–37). PCR was performed on approximately 200 ng of bisulfite-converted DNA using PyroMark PCR kits (Qiagen) on a GeneAmp PCR System 9700 (Applied Biosystems, Grand Island, NY, USA). Pyrosequencing was carried out on a PyroMark Q24 system (Qiagen) and data was analyzed using PyroMark Q24 software, version 2.0.6 (Qiagen). Bisulfite treatment controls were integrated as a quality control measure.

### 4.7. Statistical Analysis and Visualization

All figures were generated using R v4.5.1. The ggpubr package v0.6.0 (CRAN—Package ggpubr (https://www.r-project.org/)) was used to create bar graphs, box plots, and dot plots. Statistical comparisons were performed using the rstatix package v0.7.2 (CRAN—Package rstatix (https://www.r-project.org/)), which facilitated independent samples *t*-tests and the annotation of plots with significance labels. Linear regression analyses were conducted using the lm.beta package v1.7.2 (CRAN—Package lm.beta (https://www.r-project.org/)) to obtain standardized regression coefficients. Models included interaction terms to assess effect modification between CI strength or *APOC1* gene expression and variables such as disease status, *APOE* genotypes, and *APOC1* In/del genotypes. Causal mediation analyses were performed using the mediation package v4.5.1 (CRAN—Package mediation (https://www.r-project.org/)) to evaluate the indirect effects of DNA methylation on CI strength or *APOC1* gene expression levels. The assumptions of the statistical analyses, including approximate normality, were evaluated with graphical methods, specifically inspection of data distributions and Q-Q plots.

## 5. Conclusions

Our findings highlight how 3D chromatin organization at the *APOE* locus influences gene regulation relevant to AD susceptibility. Using 3C analyses to map physical interactions between key regulatory elements, we identified locus-specific architectural features and chromatin interactions involving *TOMM40*, *APOE*, and *APOC1*, underscoring coordinated regulation mediated by chromatin contacts. The convergence of histone modifications, gene annotations, and spatial interactions at the *APOE* locus supports the hypothesis that its 3D configuration actively modulates gene expression and contributes to AD risk. Moreover, genetic variants influencing *APOE* locus architecture hold promise for the development of diagnostic biomarkers and targeted therapeutics. Understanding SNP allele-specific effects on chromatin organization and gene regulation will be critical for advancing personalized medicine. Collectively, these insights provide a framework for future mechanistic studies into *APOE*’s broader role in age-related diseases, fostering an improved healthspan and lifespan in the general population.

## Figures and Tables

**Figure 1 ijms-27-00302-f001:**
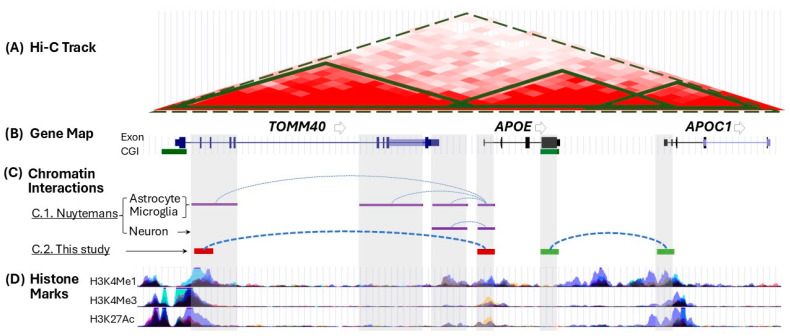
Schematic illustration of chromatin architecture at the *APOE* locus, highlighting gene-specific interaction loops and associated histone modification profiles. (**A**) Hi-C heatmap from the UCSC Genome Browser shows genomic interaction frequencies. High interaction scores (red) indicate frequent interactions. Regions with varying interaction frequencies are enclosed in solid and dashed-line triangles. (**B**) The three *APOE* locus genes are shown with their relative positions, exons, CpG islands (CGIs), and transcriptional orientations. (**C**) Chromatin interactions (blue dotted lines) between the *APOE* promoter and the *TOMM40* IVS2-5 region, based on Nuytemans et al. (purple lines) and this study (red lines). This study also discovered a novel interaction between the *APOE* exon 4 CGI and the *APOC1* promoter (green lines). (**D**) Wiggle tracks display three major histone marks in the region, as defined by the ENCODE project and visualized on the UCSC Genome Browser.

**Figure 2 ijms-27-00302-f002:**
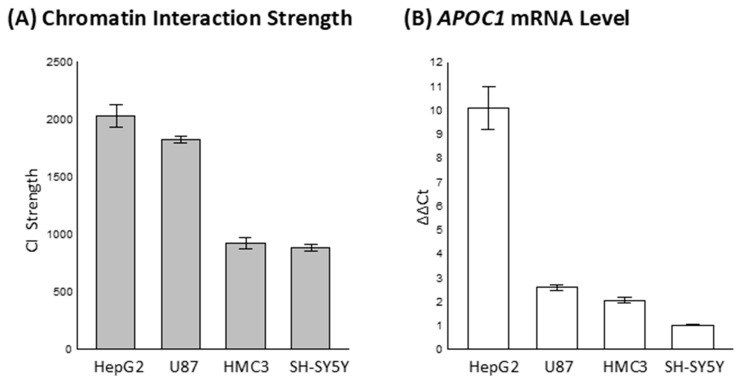
Cell type-specific chromatin interaction between the *APOE* exon 4 CpG island (CGI) locus and the *APOC1* promoter, and its relationship to *APOC1* transcription. (**A**) Chromatin interaction (CI) strengths were quantified in HepG2, U87, HMC3, and SH-SY5Y cells using the CIS-dPCR assay. (**B**) *APOC1* mRNA levels were measured by RT-qPCR (TaqMan assay) in the same cell lines. Expression was normalized using the ∆∆Ct method, where a higher ∆∆Ct value indicates greater mRNA abundance. *APOC1* ∆Ct was calculated relative to a housekeeping gene *ACTB*, and ∆∆Ct values were derived using SH-SY5Y cells as the baseline (set to 1).

**Figure 3 ijms-27-00302-f003:**
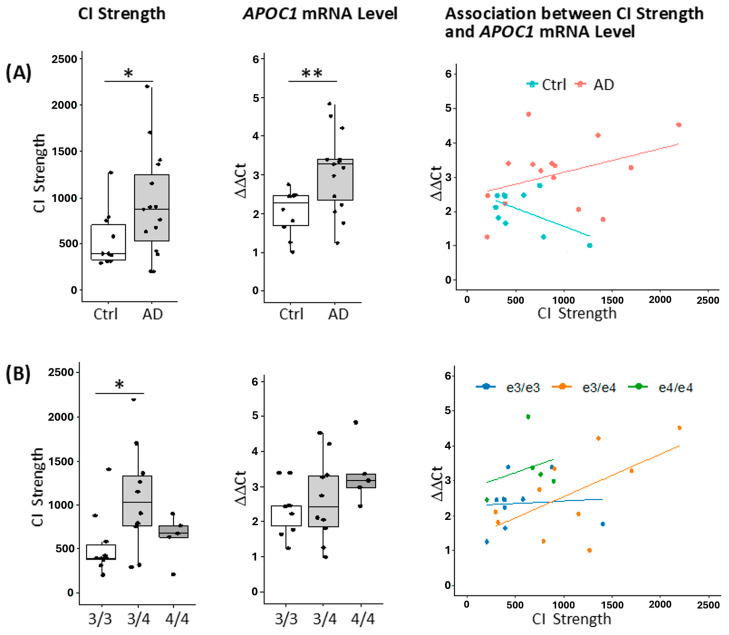
Chromatin interaction (CI) strength and *APOC1* mRNA levels in PMB samples. (**A**) Data stratified by disease status (control (Ctrl), n = 10; AD, n = 15). CIS-dPCR assay revealed increased CI strength at the *APOE*–*APOC1* locus in AD compared to in control (*p* < 0.05), accompanied by elevated *APOC1* mRNA levels (*p* < 0.005) measured by RT-qPCR using the ΔΔCt method, normalized to *ACTB* and a reference PMB sample. CI strength and *APOC1* expression showed divergent correlations in Ctrl (r = −0.56, *p* = 0.09) and AD (r = 0.39, *p* = 0.15). (**B**) Data stratified by *APOE* genotypes (ε3/ε3, n = 10; ε3/ε4, n = 10; ε4/ε4, n = 5). CI strength varied by *APOE* genotype, with highest levels in ε3/ε4 carriers (*p* < 0.05 vs. ε3/ε3). *APOC1* mRNA levels exhibited genotype-dependent trends, though not statistically significant (*p* > 0.05). Genotype-stratified correlations revealed a positive trend in ε3/ε4 carriers (r = 0.61, *p* = 0.06), with no correlation observed in ε3/ε3 or ε4/ε4 groups. An independent samples *t*-test was used to compare across the biological variables. Statistical significance is indicated as follows: *, *p* < 0.05; **, *p* < 0.005.

**Figure 4 ijms-27-00302-f004:**
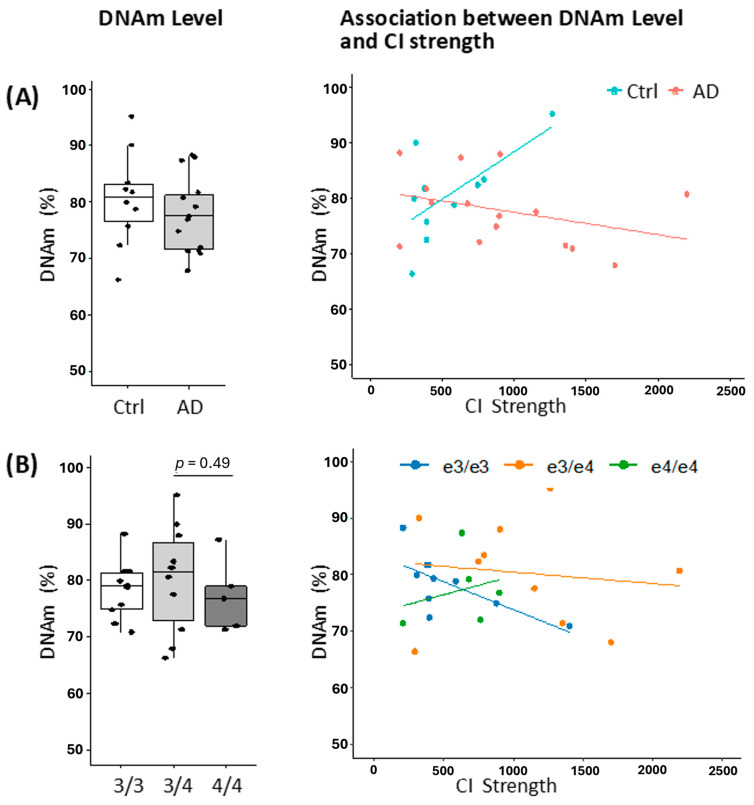
Association between chromatin interaction (CI) strength and regional DNA methylation (DNAm) levels within the chromatin loop spanning the *APOE* exon 4 CpG island (CGI) and *APOC1* in PMB tissue. (**A**) Data stratified by disease status (control (Ctrl), n = 10; AD, n = 15). DNAm levels at the *APOE*-CGI differentially methylated region I (DMR I) [60] were quantified in Ctrl and AD PMB samples. No significant group-level difference was observed (*p* = 0.383). In Ctrl samples, DNAm was positively correlated with CI strength (r = 0.65, *p* = 0.04), whereas AD samples showed a non-significant negative trend (r = −0.35, *p* = 0.20). (**B**) Data stratified by *APOE* genotype (ε3/ε3, n = 10; ε3/ε4, n = 10; ε4/ε4, n = 5). Methylation levels did not significantly differ across genotypes (*p* > 0.49). A significant negative correlation was observed in ε3/ε3 carriers (r = −0.69, *p* = 0.03), while no significant associations were found in ε3/ε4 (r = −0.13, *p* = 0.73) or ε4/ε4 (r = 0.27, *p* = 0.66) genotypes. An independent samples *t*-test was used to compare across the biological variables.

**Table 1 ijms-27-00302-t001:** Demographics of human PMB study samples.

	Ctrl	AD
Sample, *n*	10	15
Sex, Female, *n* (%)	6 (60.0)	8 (53.3)
Age at death_mean (SD)	83.8 (6.8)	84.7 (5.1)
Age at onset_mean (SD)	N/A	76.3 (6.4)
Disease duration_mean years (SD)	N/A	8.3 (4.3)
Postmortem interval_mean hours (SD)	4.8 (2.5)	4.8 (2.0)
CERAD score		
Absent	5	0
Sparse	3	0
Moderate	2	2
Frequent	0	13
BRAAK stage		
I	1	0
II	4	0
III	5	0
IV	0	0
V	0	8
VI	0	7
*APOE* genotype		
ε3/ε3	5	5
ε3/ε4	5	5
ε4/ε4	0	5

**Table 2 ijms-27-00302-t002:** Quantitative assessment of chromatin interaction strength and *APOC1* gene expression stratified by *APOE* genotype in relation to AD status.

Outcome ^a^:	Chromatin Interaction Strength	*APOC1* mRNA Level
Disease status	0.247	0.055
*APOE* ε3/ε4	0.408	0.285
*APOE* ε4/ε4	0.024	0.570
Disease and *APOE* ε3/ε4	0.427	0.651
Disease and *APOE* ε4/ε4	NA	NA

Linear regression analyses with interaction terms were conducted. Models include covariates such as disease status, *APOE* genotype, and *APOC1* genotype. ^a^ Effect size is presented as the standardized regression coefficient. NA indicates unavailable data.

**Table 3 ijms-27-00302-t003:** Assessment of the relationship between regional DNA methylation and locus-specific chromatin interaction strength or *APOC1* gene expression.

Outcome ^a^:	Chromatin Interaction Strength		*APOC1* mRNA Level	
	Estimate [95% CI]	*p*-Value	Estimate [95% CI]	*p*-Value
ACME	−7.10 [−123.82, 114.01]	0.934	−0.021 [−0.220, 0.219]	0.874
ADE	345.77 [−16.00, 678.07]	0.066	0.383 [−0.328, 1.165]	0.278
Total Effect	338.67 [18.82, 674.22]	0.042	0.362 [−0.322, 1.166]	0.298
Proportion Mediated	−0.02 [−0.71, 0.50]	0.916	−0.057 [−2.179, 1.497]	0.916

^a^ Causal mediation analysis was performed to evaluate indirect effects of DNA methylation on chromatin interaction strength and *APOC1* gene expression. This model included covariates for disease status, *APOE* genotype, and *APOC1* genotype.

## Data Availability

The original contributions presented in this study are included in the article/Appendix A. Further inquiries can be directed to the corresponding author.

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
