# Peer review of "Int. J. Mol. Sci.2026, 27(1), 302;https://doi.org/10.3390/ijms27010302"

_ijms, 2025, doi:10.3390/ijms27010302_

Round 1

Reviewer 1 Report

Comments and Suggestions for Authors

The authors here reported the use of chromatin interaction strength to evaluate the APOE-linked disease risk. The linkage between APOE variant and APOC1 in regulating the interaction and enhancing APOC1 transcription in Alzheimer’s disease (AD) samples has been well consolidated by the chromatin conformation capture (3C). It will further confirm the role of APOE locus as the risk factor in predicting potential AD susceptibility.This work is very meaningful and will stimulate more valuable thoughts about how to perform the sophisticated diagnosis and therapy with APOE as the target.Also, I have some suggestions for further improvement of this manuscript to facilitating the reading.

1) The coordinated regulation among genes at the APOE locus including TOMM40, APOE, and APOC1 is not fully demonstrated. Except Figure S1, there is no any results about the chromatin interactions spanning the TOMM40–APOE region and the TOMM40 mRNA transcription as presented in Figure 2, Figure 3, Figure 4 only focusing on APOC1. Please also provide the explanations why most work focuses on APOE-APOC1 interactions.

2) For a direct understanding by readers, the hypothesis that its 3D configuration plays an active role in modulating gene expression and contributes to AD riskcan be illustrated with a model figure.

3) The resolution of Figure 1 can be further improved.

4) The discussion is too long. You can add some subtitles to make easier reading.

Reviewer 2 Report

Comments and Suggestions for Authors

That’s interesting research. I have some basic questions on your study ?
What evidence supports causality rather than correlation between 3D chromatin remodeling and transcriptional changes ?
Which restriction enzymes were used in the 3C protocol, and how was digestion efficiency validated across samples?
Did you test the multiple viewpoints within the locus to confirm interaction reciprocity?
APOC1 transcription was compared across genotypes and AD status. Which normalization method was used ?
The key conclusion is not clearly defined, either in the abstract or within the main manuscript. Please revise it to ensure that the central takeaway of the study is explicitly stated and easily understood by readers.
The introduction would benefit from a more comprehensive review of previous studies about relationship between APOE and TOMM40 and others.
Please have the manuscript reviewed for grammatical accuracy by a native speaker.
Please specify the statistical tests used to assess the normality of the variables.
In the introduction, some sentences are long and hard to follow.
The first paragraph of the discussion should summarize all key findings rather than simply repeating the results; please revise accordingly.
The manuscript is generally comprehensible, but the English language requires moderate editing for clarity and style. There are numerous grammatical errors, inconsistent tense usage, and some awkward phrasing.
The introduction provides basic background but lacks a critical review of existing literature regarding APOE role during diseases occurred. There are several important references that are not cited.
The results are internally consistent but should be interpreted with caution due to the methodological weaknesses outlined above.
The discussion section needs to include more detailed information.
The discussion should not simply restate results. Please interpret your findings in the context of other studies.
The abstract does not mention properly the aim of the study.
The methods section in the abstract is incomplete or unclear.
The results in the abstract are not clearly or adequately described.
The abstract needs to be completely rewritten.

Round 2

Reviewer 1 Report

Comments and Suggestions for Authors

This resubmitted manuscript has been well reivsed according to reviewer's suggestions. I recommend its publication in IJMS in its present form. 

Author Response

No additional comments from the reviewer 1 were provided.

Reviewer 2 Report

Comments and Suggestions for Authors

Please check the table front size. 

Author Response

In response to the reviewer’s comment, Tables 1, 2, and 3 have been reformatted in Word format. The revised tables are provided in the attachment.
